# GSK3β: A Master Player in Depressive Disorder Pathogenesis and Treatment Responsiveness

**DOI:** 10.3390/cells9030727

**Published:** 2020-03-16

**Authors:** Przemysław Duda, Daria Hajka, Olga Wójcicka, Dariusz Rakus, Agnieszka Gizak

**Affiliations:** Department of Molecular Physiology and Neurobiology, University of Wrocław, Sienkiewicza 21, 50-335 Wrocław, Poland; daria.hajka@uwr.edu.pl (D.H.); olga.wojcicka@uwr.edu.pl (O.W.); dariusz.rakus@uwr.edu.pl (D.R.)

**Keywords:** GSK3β, MDD, depression, anti-depressants, AKT, BDNF, neuroprotection

## Abstract

Glycogen synthase kinase 3β (GSK3β), originally described as a negative regulator of glycogen synthesis, is a molecular hub linking numerous signaling pathways in a cell. Specific GSK3β inhibitors have anti-depressant effects and reduce depressive-like behavior in animal models of depression. Therefore, GSK3β is suggested to be engaged in the pathogenesis of major depressive disorder, and to be a target and/or modifier of anti-depressants’ action. In this review, we discuss abnormalities in the activity of GSK3β and its upstream regulators in different brain regions during depressive episodes. Additionally, putative role(s) of GSK3β in the pathogenesis of depression and the influence of anti-depressants on GSK3β activity are discussed.

## 1. Introduction

### 1.1. Major Depressive Disorder

According to the World Health Organization’s statistics for March 2018, depression is a major contributor to the overall global burden of diseases. Globally, more than 300 million people suffer from depression. In the Diagnostic and Statistical Manual of Mental Disorders (DMS-5), major depressive disorder (MDD), the principal form of depression, is characterized by the following key symptoms: depressed mood and anhedonia (which are the fundamental symptoms), suicidal ideation, plan or attempt, fatigue, sleep deprivation, loss of weight and appetite, and psychomotor retardation [1]. The first effective treatment for MDD was established in the 1950s when anti-depressant effects of iproniazide and imipramine were discovered. Iproniazide, originally described as an anti-tuberculosis drug, was found to be the first monoamie oxidase inhibitor (MAOi), whereas imipramine, an anti-histamine drug, was studied as an antipsychotic for use in patients with schizophrenia [2,3]. Latter, imipramine became one of the first members of tricyclic anti-depressants (TCAs). These findings resulted in the monoaminergic hypothesis of the MDD pathogenesis, which assumes that MDD is caused by a reduction in noradrenaline (NA) and serotonin (5-HT) neurotransmission [4]. In the current pharmacotherapy of depression, three main groups of drugs are available: TCAs, the first generation of anti-depressants, selective serotonin reuptake inhibitors (SSRIs), and selective serotonin and noradrenaline reuptake inhibitors (SSNRIs, the second generation). In addition, medications, such as α2-receptor blockers, MAOis, selective noradrenaline reuptake inhibitors (SNRIs), selective noradrenaline and dopamine reuptake inhibitors (SNDRIs), melatonin receptor agonists and serotonin 5-HT_2C_ receptor antagonists, are used in the treatment of depression. Some anti-depressants, such as trazodone, do not belong to any of the groups listed above. Additionally, electroconvulsive therapy, conducted for the first time in 1938, is still widely used in the treatment of MDD, especially in its drug-refractory form [5].

Although the monoaminergic hypothesis has led to the invention of many successful therapeutic strategies based on the elevation of levels of NA and 5-HT in the synaptic cleft, it does not explain the anti-depressant effect of lithium and the rapid action of ketamine in the treatment of mood disorders [6]. Therefore, factors other than neurotransmission must be taken into consideration in the context of the MDD pathogenesis. One of them is glycogen synthase kinase 3β (GSK3β) signaling.

### 1.2. Glycogen Synthase Kinase 3β

GSK3 was isolated in 1980, from rabbit skeletal muscle, and described as a highly specific serine/threonine kinase for glycogen synthase [7]. There are two isozymes of GSK3, α and β, and both are expressed at similar levels in the mouse brain [8]. In the human brain, the β isozyme predominates [9]. Therefore, GSK3β is expected to be crucial for the human central nervous system functioning. The activity of GSK3β is regulated positively and negatively by phosphorylation on Tyr216 and Ser9, respectively [10,11]. Whereas phosphorylation of the residue Tyr216 occurs during the GSK3β translation process and results in a synthesis of the fully activated kinase, Ser9 phosphorylation seems to be the main regulatory modification during the enzyme lifespan [12]. Ser9-phosphorylated GSK3β remains inhibited, and dephosphorylation of the residue results in the disinhibition (activation) of the kinase.

GSK3β is part of numerous cellular signaling pathways, and its activity can be regulated, directly or indirectly, by several kinases, phosphatases, and proteases. The wide spectrum of GSK3β substrates, including transcription factors, glycolytic enzymes, pro- and anti-apoptotic factors, mitochondrial channels, membrane receptors, and cytoskeleton-associated proteins, makes GSK3β a central point of the cell homeostasis maintenance [13]. The activity of GSK3β affects energy metabolism, cell survival, proliferation, apoptosis, membrane polarity, internalization of the synaptic receptors, neuroplasticity, neurotransmission, amyloid processing, and many other processes [13]. 

Extracellular factors, such as insulin or insulin-like growth factor 1 (IGF-1), epidermal growth factor (EGF), platelet-derived growth factor (PDGF), and transforming growth factor 1β (TGF-1β), acting via the phosphoinositide 3-kinase/protein kinase B (PI3K/AKT) pathway, inhibit GSK3β by phosphorylation of Ser9 of the kinase [14,15,16]. The same effect is mediated by mitogen-activated protein kinase/extracellular signal-regulated kinase (MAPK/ERK) pathway activity upon EGF, fibroblast growth factor (FGF), PDGF, nerve growth factor (NGF), and brain-derived neurotrophic factor (BDNF) stimulation, and as a result of cytokine action via p38 MAPK [17,18,19]. Inactivation of GSK3β may also be an effect of protein kinase A (PKA) activity (stimulated by. e.g., elevated cAMP level), integrin-linked kinase (ILK), calcium/calmodulin dependent protein kinase 2 (CaMK2) and WNT signaling [16,20,21,22]. On the other hand, dephosphorylation of Ser9 by protein phosphatase 1 (PP1), protein phosphatase 2A (PP2A), and protein phosphatase 2B (PP2B, calcineurin) directly activates GSK3β [23]. It is noteworthy that the majority of the proteins mentioned above are involved in intracellular processes related to neurotransmission and neuroplasticity.

### 1.3. GSK3β Activity in Neurotransmission and Neuroplasticity

Glutamate, the most abundant excitatory neurotransmitter in the vertebrate central nervous system, acts through α-amino-3-hydroxy-5-methyl-4-izoxazole propionic acid receptors (AMPAR) and *N*-methyl-d-aspartate receptors (NMDAR). Whereas AMPARs are responsible for membrane depolarization upon glutamate binding, NMDARs, conducting calcium current, are directly associated with neuroplasticity [24]. The activity of NMDARs results in a high cytosolic calcium concentration ([Ca^2+^]_c_) and leads to calcium/calmodulin-dependent protein kinase 2 (CaMK2) activation. In contrast, modest [Ca^2+^]_c_ causes calcineurin activation [25]. Both enzymes are known to modify Ser9 residue of GSK3β. CaMK2 stimulates its phosphorylation and thus, inhibits GSK3β, while calcineurin induces its dephosphorylation and activates the kinase [22,26]. A change in cellular [Ca^2+^]_c_ underlies the NMDAR-dependent neuroplasticity phenomena called long-term potentiation (LTP) and long-term depression (LTD). LTP is a process of strengthening of synapses caused by NMDAR-activation-evoked high [Ca^2+^]_c_. Calcium ions activate CAMKs leading to the incorporation of AMPARs into the postsynaptic membrane and synthesis of new subunits of AMPAR [27]. In contrast to LTP, LTD is a process of weakening of synapses in which AMPARs are internalized due to the modest-[Ca^2+^]_c_-induced activation of protein phosphatases, including calcineurin. As a result, dephosphorylated (and activated) GSK3β phosphorylates postsynaptic density protein 95 (PSD95) and kinesin light chain 2 which regulates AMPARs internalization [28].

To summarize, active GSK3β is related to the downsizing of synapses and decreased excitability of neurons, whereas inhibition of GSK3β is necessary for the induction of LTP, the process underlying new memory formation.

γ-aminobutyric acid (GABA) is a main inhibitory neurotransmitter in the central nervous system. Its receptors, ionotropic GABA_A_ and metabotropic GABA_B_ receptors, group in inhibitory synapses together with gephyrin, a scaffolding protein [29]. GSK3β phosphorylates gephyrin, which induces the formation of new GABAergic synapses [30]. On the other hand, GABAergic transmission inhibits GSK3β by acting through GABA_B_ receptors, which stimulate the activity of AKT in a β-arrestin 2-dependent manner [31].

Another neurotransmitter that can modify the activity of GSK3β is dopamine (DA). DA is involved in motor control, motivation, reward, and executive functions. The effects of its signaling on GSK3β depend on the type of DA receptors expressed on the neuronal surface. Stimulation of Dopamine Receptor 1 (D1R) and Dopamine Receptor 3 (D3R) activates adenylyl cyclase and inhibits GSK3β through PKA and AKT action [32]. Dopamine Receptor 2 (D2R) inhibits AKT via activation of β-arrestin 2/PP2A complex and leads to GSK3β activation [33].

GSK3β activity can also be affected by 5-HT signaling. Stimulation of 5-HT_1_ and 5-HT_7_ activates the PI3K/AKT pathway and thus, increases Ser9 phosphorylation in GSK3β, while activation of 5-HT_2A_R has the opposite effect [34,35].

NA inhibits GSK3β activity acting through the α1A-adrenergic receptor (α1AAR) and stimulating phosphorylation of GSK3β Ser9 via protein kinase C (PKC) [36], and through α2- and β-adrenergic receptors (α2AR and βAR) via PKA [37,38].

The relationship between GSK3β and neurotransmitters is schematically presented in Figure 1.

Concluding, GSK3β is a hub that links different molecular pathways within a cell. The activity of the kinase, affected by the action of neurotransmitters, mediates neuroplasticity and directs neurons towards synaptic potentiation or depression route. Thus, because MDD is believed to be a result of dysregulation in neurotransmitters actions in different brain regions, GSK3β can be considered as a factor engaged in the MDD pathogenesis and development.

## 2. GSK3β Expression Profile and Activity in Depression

Expression profile, haplotypes, and activity of GSK3β have been intensively studied in a broad range of psychiatric and neurological diseases, such as schizophrenia, Parkinson’s, and Alzheimer’s diseases [13], and in the context of mood disorders [39]. It has been found that the total amount of GSK3β protein is unchanged, whereas its activity is significantly enhanced in the prefrontal cortex (PFC) of depressed patients and depressed suicide victims, but not in suicides without a psychiatric history. This suggests a role of GSK3β in MDD but not in suicide per se [40,41].

The increase on GSK3β activity has also been found in platelets of depressed patients [42,43], whereas the GSK3β gene expression was upregulated in such brain structures as the frontal cortex, raphe, and hippocampus in the rat model of depression [44].

Anhedonia and loss of motivation are some of the main MDD symptoms. They are caused by a disrupted interplay between different brain structures, such as ventral tegmental area, nucleus accumbens, and cingulate cortex, which can be grouped in the so-called reward circuit [45]. Wilkinson et al. demonstrated that in the nucleus accumbens, a key region of the circuit, the amount of Ser9-phosphorylated GSK3β is downregulated in the mouse social defeat model of depression [46]. The increased activity of the kinase can be observed in susceptible, but not in resilient animals. Additionally, a similar pro-depression-like effect can be induced by GSK3β overexpression in the nucleus accumbens, while the expression of inactive GSK3β mutant promotes resilience to social defeat stress [46]. On the other hand, Crofton et al., found that the knockdown of GSK3β in the nucleus accumbens shell increases cocaine self-administration and depression-like behavior in social contact tests in rats [47,48]. This discrepancy between stressor type and its effect within the same structure is also reflected in the activity of the ventral tegmental area, a midbrain structure that delivers input to nucleus accumbens. It has been shown that chronic social defeat stress enhances the phasic firing rate of the ventral tegmental area neurons in defeated rodents [49]. However, chronic stress can also lead to atrophy of the ventral tegmental area system [50]. It has been hypothesized that these opposing outcomes might be due to a different nature of the stressors, the heterogeneity of ventral tegmental area cells, or the time of day when the experiments were performed [51]. Interestingly, it has been found that the level of Ser9-phosphorylated GSK3β in the ventral tegmental area and other reward-related brain structures in naïve rats shows significant circadian rhythmicity [52].

The activity of AKT, a negative upstream regulator of GSK3β, is decreased in the prefrontal cortex of depressed suicide victims [40]. In line with this, decreased AKT activity in the ventral tegmental area increases susceptibility to depressive-like behaviors in a rodent model [53].

Interestingly, different haplotypes of GSK3β seem to be related to MDD severity, age of onset, and drug responsiveness. Single nucleotide polymorphism (SNP) rs6782799 in GSK3β gene has been demonstrated to be important for susceptibility to MDD by modification of the relationship between negative life events and depression [54,55]. Another SNP, rs334555, alters the age of MDD onset. Its C/C and C/G genotypes are associated with the late age of onset (about 46 and 41 years old on average, respectively), whereas its G/G genotype with the early (about 28 years old on average) onset of depression [56]. Although rs334555 has no direct effect on GSK3β, it is in a moderate linkage disequilibrium with rs6438552, which is an intronic SNP related to the altered splicing and increased level of the GSK3β transcript in lymphocytes [57].

rs6438552 has been associated with a lowered volume of grey matter both in right and in left superior temporal gyri, and right hippocampus of MDD patients [58]. Additionally, rs6438552, together with rs334558, has been linked to anxiety-like behavior in MDD [59]. rs334558, a promoter SNP related with enhanced GSK3β transcription [57], is associated with remission upon anti-depressant drugs administration [60].

Summarizing, the activity of GSK3β in different brain regions is affected by stress, and haplotypes of the kinase determine a severity, age of onset, and drug responsiveness in MDD.

## 3. Putative Role of GSK3β in the MDD Pathogenesis

The current pharmacotherapy of MDD originates from the monoaminergic hypothesis. However, the MDD pathogenesis appears to be more complex. Various animal models of depression have been established: olfactory bulbectomy, learned helplessness, maternal separation, social isolation, chronic unpredictable/mild stress, witness defeat, and many others. None of them fully recapitulated the entire human depression syndrome [61]. This suggests that the MDD onset engages many cellular processes and factors. Most of them can be directly or indirectly linked to altered GSK3β activity. The place of GSK3β in different molecular pathways is schematically presented in Figure 2.

### 3.1. GSK3β in the Animal Models of Depression

In 1949, Cade suggested that lithium, the classical mood stabilizer, might be a possible therapeutic for psychiatric diseases based on its behavioral effects in guinea pigs [62]. Since then, the rodent behavioral characteristics have become the high-value models in mood disorders treatment studies [39]. The discovery that GSK3β is a direct target of lithium action [63] has raised the possibility that lithium exerts its effects through the modulating activity of the kinase. To identify the mood-altering role of GSK3 α and β isozymes, molecular methods have been employed in various animal models of depression.

On the one hand, behavioral characteristics of GSK3α/β_21A/21A/9A/9A_ knock-in mice (with serine to alanine mutations to block inhibitory phosphorylation of serine 21 and 9 in GSK3 α and β, respectively) demonstrated that the animals exhibited a heightened response to a novel environment and that administration of amphetamine causes over 2.5-fold greater hyperactivity compared to control mice [64]. It emphasized the importance of GSK3α/β activity in the development of manic-like disturbances. On the other hand, the knock-in animals showed increased vulnerability to stress-induced depressive-like behavior in the learned helplessness, forced swim, and tail suspension tests [64]. Additionally, the lack of adaptability of these knock-in mice to stress involved also anxiety, which often coexists with depression. The animals displayed a mild-anxious behavior in the elevated plus maze [64]. Later, using GSK3β knock-in mice, it has been established that increased activity of the β isozyme of GSK3 is sufficient to impair mood regulation in learned helplessness model of depressive-like behavior, whereas GSK3α activity alone does not impair this process [65].

Bilateral intra-hippocampal injections of lentivirus expressing shRNA anti-Gsk3β induce an antidepressant-like effect in chronically stressed mice in the forced swim and tail suspension tests [66]. Moreover, heterozygous loss of *Gsk3β* causes behavioral defects that mimic the action of lithium [67], whereas transgenic expression of *Gsk3β* in *Gsk3β*^+/−^ heterozygotes reverses these defects [68]. Additionally, the same effect is observed when *GSK3β* is overexpressed in lithium-treated mice [68]. Rescue of the heterozygous loss of *Gsk3β* or lithium-induced phenotype by restoring the activity of GSK3β strongly supports the hypothesis that the phenotype is due to specific inhibition of the kinase.

### 3.2. Inhibitors of GSK3β in Depression

Several lines of evidence have shown that GSK3β contributes to the development of such prevalent diseases as diabetes, Alzheimer’s disease, as well as mood disorders. GSK3β inhibitors can be classified into three categories: non-ATP-competitive, ATP-competitive, and substrate competitive inhibitors [69].

It has been demonstrated that non-ATP-competitive GSK3β inhibitors ameliorate depressive-like behavior in rodents. It has also been shown that prolonged learned helplessness is reversible and is maintained by abnormally active GSK3, whereas treatment with TDZD-8, non-ATP-competitive GSK3 inhibitor, reverses the impaired recovery from learned helplessness [70]. In turn, an ATP-competitive GSK3β inhibitor SB216763 has been found to increase anti-depressant responses in the forced swim test [71], whereas SAR502250 improved the stress-induced physical state in the chronic mild stress test in mice [72].

Intracerebroventricular injection of a novel GSK3β substrate competitive inhibitor, L803-mts, has reduced the duration of immobility in the forced swim test in mice, in comparison to control animals [73]. Additionally, the expression level of β-catenin, a substrate of GSK3β, was increased in the inhibitor-treated animals [73].

The above studies demonstrate that GSK3β inhibitors produce anti-depressive-like behavior and suggest the potential of the kinase inhibitors as anti-depressants.

### 3.3. BDNF-Regulated Action of GSK3β

In 1997, a reduction in the cortical volume of patients suffering from different types of depression was linked to decreased brain activity [74]. Since then, lowered cell density in PFC and decreased cortical thickness in patients with depression have been demonstrated [75]. Moreover, depressive behavior in rats is related to persistent remodeling of hippocampal synapses [76] and dendritic atrophy in hippocampi [77]. The decreased volume of the hippocampus has also been found in MDD patients in comparison to healthy individuals in an MRI-based study [78]. Similar effects are observed in rat and mouse medial PFC [79,80]. In contrast to the hippocampus and PFC, dendritic arborization and dendritic spines number are increased within limbic regions, such as nucleus accumbens and amygdala [81,82]. These changes are correlated with anxiety-like behavior and anhedonia [83]. Changes can also be observed in other brain regions, such as the ventral tegmental area [84]. Additionally, in the adult rodent brain, repeated stress reduces the neurogenesis ratio and the total neurons number in the dentate gyrus [85].

All the above changes might be linked to the altered activity of BDNF, a neurotrophic factor that supports the survival of existing neurons and improves the growth of new neurons and synapses [86]. However, the effect of BDNF on forebrain and mesolimbic circuitry during depressive episodes is heterogeneous. In a rat model of depression, the concentration of BDNF is reduced in the hippocampus [87], and the release of BDNF seems to mediate the rapid action of ketamine in PFC [88]. On the other hand, the levels of BDNF in structures of the mesolimbic system, such as nucleus accumbens and ventral tegmental area, are increased due to chronic stress [89], which promotes a pro-depressant phenotype. It corresponds to results of post mortem studies of human brains where an increased level of BDNF in nucleus accumbens has been detected [90].

The regulatory effect of BDNF on GSK3β activity is well characterized. BDNF is a ligand for tropomyosin receptor kinase B (TrkB), the stimulation of which leads to the activation of PI3K/AKT and ERK1/2 signaling pathways [91]. Both of these pathways reduce the activity of GSK3β, but only PI3K/AKT acts through phosphorylation of Ser9 [92,93]. Thus, factors influencing the BDNF level in the brain also impact GSK3β activity.

In contrast to the reduced level of BDNF in the PFC and hippocampus [87], the concentration of the factor is elevated in mesolimbic structures in depression [89]. Simultaneously, atrophy of the PFC and hippocampus, and increased synaptogenesis in limbic regions, are observed in depressed individuals [75,77,81,82]. This might be related to the BDNF-dependent GSK3β activity reduction. The Ser9 of GSK3β is highly phosphorylated upon stimulation of synaptogenesis, and the inhibition of the kinase is required for dendritic growth and arborization, whereas an increase in its activity leads to marked shrinkage of dendrites [94,95]. Moreover, in vivo overexpression of GSK3β reduces neurogenesis in adult hippocampus [96] and induces pro-depressant-like events [97].

One of the downstream targets of BDNF is p11 (also called S100A10), the expression of which is positively regulated by BDNF through TrkB and via the MAPK/ERK signaling pathway [98]. p11 is a calcium effector protein. It modulates signal transduction associated with serotonin receptors (especially 5-HT_1B_R) stimulation [99] and increases plasma membrane localization of 5-HT_1B_Rs [100]. p11 is downregulated in depressed patients [100], and overexpression of p11 has the anti-depressant-like effect [100]. Furthermore, p11 knockout mice exhibit a pro-depressant phenotype and are insensitive to the anti-depressant action of BDNF [98]. This might be correlated with a lowered amount of membrane-localized 5-HT_1B_Rs, which mediates inhibition GSK3β via AKT activation [34].

BDNF induces protein synthesis via activation of the mammalian target of rapamycin (mTOR), which can be inhibited by REDD1 (regulated in development and DNA damage response-1) protein. The expression of REDD1 is elevated in PFC due to chronic stress [101]. Post mortem studies of PFC tissue from patients suffering from depression have shown an increase in REDD1 [101] and a reduction in mTOR protein levels [102]. Phosphorylation of REDD1 leads to its proteasomal degradation and to the recovery of mTOR signaling [103]. Interestingly, REDD1 may be phosphorylated by GSK3β, which triggers the recruitment of the E3 ligase complex and results in REDD1 ubiquitination and degradation by the proteasome [103]. This unexpected action of GSK3 in line with BDNF might be explained by the enhanced expression of ATF4 upon endoplasmic reticulum (ER) stress conditions. In such conditions, neuronal ATF4 can be upregulated (see below), which increases the expression of REDD1 [104] and simultaneously reduces GSK3β inhibition [105].

### 3.4. GSK3β and the Unfolded Protein Response

During the last years, several lines of evidence have demonstrated a strong connection between depression and unfolded protein response (UPR). UPR is a cellular stress response mechanism activated upon the accumulation of misfolded proteins in the ER. Activation of UPR leads to restoration of the ER homeostasis or if it cannot be achieved, to apoptosis [106]. The apoptosis is promoted by activation of the ATF4 and ATF6 transcription factors, which, in turn, induce expression of the pro-apoptotic C/EBP homologous protein (CHOP) [107,108].

The signs of ER-stress-induced UPR have been found in brains of depressed patients who died by suicide [109]. This strengthens the connection between depression and UPR. Moreover, UPR activation has been observed in several tauopathies, where the involvement of GSK3β in tau phosphorylation is well established [110]. As a result, the role of the kinase in ER-stress-induced UPR has been intensively investigated. Under stress conditions, the pro-apoptotic CHOP protein is regulated by GSK3. However, GSK3 inhibitors affect neither ATF4 nor ATF6 activity, which suggests that GSK3/CHOP interaction might be another ATF-independent factor in the life/death switch mediated by UPR [105]. In line with this, after UPR activation, the level of GSK3β phosphorylated on Ser9 is diminished [105].

It has been shown that UPR may be active also in mitochondria of the murine model of depression [111], which points to a new mechanism involved in the development of this disease.

Taking into account the results of the above studies, GSK3 appears to be strongly connected with the UPR, which is one of the potential causes of depression.

### 3.5. β-Catenin Destruction Complex

β-catenin, a multifunctional protein downstream to WNT signaling [112], is another factor putatively regulating stress resilience development [113]. WNT acts through membrane protein Frizzled and activates casein kinase 1 (CK1) and Disheveled (DSH). They, in turn, disrupt the so-called β-catenin destruction complex and activate β-catenin. This positively regulates the expression of WNT-related genes [46].

The β-catenin level is decreased in PFC tissue samples from MDD patients [41]. This is in line with an observation that overexpression of β-catenin mimics the effect of lithium [114]. GSK3β is a part of the β-catenin destruction complex, and its activity regulates β-catenin action. GSK3β phosphorylates and activates other members of the β-catenin destruction complex, axin, and adenomatous polyposis coli (APC). This results in GSK3β-dependent phosphorylation of β-catenin and leads to its subsequent proteasomal degradation [115]. In the presence of the WNT signal, the activity of GSK3β is inhibited due to the disruption of the β-catenin destruction complex [46]. Additionally, upon WNT activation, a Frizzled-associated protein—low-density lipoprotein receptor-related protein 5 and 6 (LRP5/6), exposes its GSK3β pseudo-substrate motif (Pro-Pro-Pro-Ser-Pro-X-Ser) and inhibits GSK3β in a competitive manner [116].

### 3.6. GSK3β-miRNA Interaction

β-catenin regulates the expression of various miRNA species, which play a role in neuroplasticity [117]. The first step of miRNA maturation occurs in the nucleus and engages the ribonuclease III enzyme called Drosha. GSK3β phosphorylates Drosha on Ser300 and Ser302 residues. This is required for the nuclear translocation of Drosha [118]. Inactivation of GSK3β results in the inhibition of maturation of the miR-302-367 cluster activated by WNT/β-catenin signaling, and miR-181 family [119]. On the other hand, miR16 and miR135a downregulate GSK3β expression (according to Mouse Genome Informatics Scientific Curators, MGI Ref. ID J:208678). Both the miRs are decreased in the blood of MDD patients [120], which corresponds to an increased level of GSK3β [39]. Moreover, miR16 regulates the expression of 5-HT transporter (SERT) [121], while miR135a is associated with the regulation of the expression of SERT and 5-HT_1A_R [122]. In MDD patients, the expression of miR135a increases after the implementation of anti-depressants (TCAs). This suggests the anti-depressant effect of miR135a [122].

### 3.7. Role of GSK3β in DNA Methylation

Some individuals suffering from MDD exhibit hyperactivity of the hypothalamic-pituitary-adrenal axis (HPA), which may be caused by increased production of the corticotrophin-releasing factor (CRF) [123]. An increased number of CRF-neurons has been found in the paraventricular nucleus of depressed patients [124]. CRF is probably involved in vulnerability to mood disorders in animal models of depression, which revealed a decreased methylation of CRF promoter upon social stressors [125]. Such a decrease in DNA methylation is in line with the altered expression of DNA methyltransferase 3a (DNMT3a) in the brain structures of depressed animals [126]. Moreover, an increased risk of suicide in humans is correlated with hyper-methylation of the BDNF promoter [127]. The expression of DNMT3a is regulated by c-Myc, a target of GSK3β [128]. The inhibition of GSK3β enhances the transcriptional activity of c-Myc. This upregulates the expression of DNMT3a [129]. Thus, the increased activity of GSK3β may be correlated with the lowered DNA methylation.

### 3.8. Neuroinflammation in Depression

Anti-inflammatory drugs may have anti-depressant effects in MDD patients [130,131]. Systemic injection of pro-inflammatory cytokines induces depression-like phenotype [132]. The elevated cytokine levels may be normalized by an anti-inflammatory treatment [133]. Depression-like behavior in rodents correlates with increased levels of pro-inflammatory cytokines: interleukin-1β (IL-1β), interleukin-6 (IL-6), and tumor necrosis factor α (TNFα) [133,134], and decreased levels of anti-inflammatory interleukin-10 (IL-10) [135]. Nuclear factor κ-light-chain-enhancer of activated B cells (NF-κB) promotes depression-like behaviors and inhibits neurogenesis in the hippocampus upon TNFα receptor (TNFR) and IL-1 receptor (IL-1R) activation due to chronic stress [136]. Elevated cytokine levels are accompanied by microglia activation/hyper-reactivity in PFC, cingulate cortex, and insula of depressed patients [137,138]. Pro-inflammatory cytokines, such as IL-6, can also be produced by astrocytes, and this can inhibit neurogenesis in the hippocampus [139]. On the other hand, cytokines alter astrocyte signaling, function, and amount. Post mortem studies have revealed a reduction in glial cell density in PFC, amygdala, and hippocampus of depressed patients [140]. This has been confirmed by studies showing decreased expression of the glial fibrillary acidic protein (GFAP, an astrocytic cell marker) in the PFC of depressed patients [141]. Astrocytic end feet, together with endothelial cells and pericytes, form the blood-brain barrier (BBB). Its integrity can be affected by malfunctions of astrocytes. This leads to increased permeability of the BBB and enhanced monocyte trafficking from the bloodstream to the central nervous system, which might strengthen the inflammatory reaction in the brain [142]. Additionally, it has been shown that IL-6 induces the production of inflammatory T helper 17 cells (Th17), thus increasing levels of these cells in the brain during depression-like states [143]. Moreover, it has been shown that the administration of Th17 cells promotes depression-like behaviors in mice, and inhibition of production and functioning of Th17 cells reduces the vulnerability of the animals to depression-like behavior [143].

NF-κB is a protein complex indispensable for the expression of inflammation-related genes and, thus, for the induction of inflammation. In unstimulated cells, NF-κB has cytosolic localization due to the activity of IκBα (nuclear factor of kappa light polypeptide gene enhancer in B-cells inhibitor, alpha) which binds to NF-κB and masks its nuclear localization signal [144]. Stimulation of TNFR and IL-1R activates IκB kinase (IKK), which inhibits IκBα. This leads to the translocation of NF-κB to the nucleus, where it can act as a transcription factor [145].

The role of GSK3β in the modulation of the inflammatory response is as complex as the inflammatory transduction signal pathway itself [146].

The expression of IL-6 is regulated by a transcription factor STAT3 [147]. GSK3β promotes STAT3 activation and, thus, stimulates the expression of IL-6. Inhibition and knockdown of GSK3β, but not GSK3α, strongly inhibits IL-6 production by glial cells both in vitro and in vivo [147].

GSK3β phosphorylates two members of NF-κB complex: p65 (RelA) and p105 (NF-κB1) [148,149]. This results in enhancement of the transactivation potential of p65 and prevents the proteasomal degradation of p105 in unstimulated cells [148]. However, upon TNFα stimulation, the GSK3-phosphorylated p105 undergoes subsequent phosphorylation by IKK, which leads to p105 degradation [148]. Thus, GSK3β plays a dual role in p105 stabilization, depending on whether or not the cells are stimulated. GSK3β also phosphorylates a transcriptional co-activator of NF-κB p50 homodimer: B-cell lymphoma 3-encoded protein (BCL-3) [148,149]. This leads to the degradation of BCL-3 and, therefore, the reduction in NF-κB activation [148]. Summarizing, GSK3β can either favor a rapid NF-κB activation or limit the activity of the factor. The effect of the kinase action depends on the activated pathway. 

Elevation of cytokines and chemokines levels in the hippocampi of mice displaying depression-like behavior is mediated by Toll-like receptor 4 (TLR4) activity [150]. It has been shown that the learned helplessness paradigm activates GSK3 in a wild-type mouse hippocampus, but not in TLR4 knockout mice [150]. Additionally, TDZD-8 attenuates an increased activation of NF-κB upon TLR4 stimulation [150], which indicates that GSK3 mediates a TLR4-related pro-inflammatory reaction associated with depression-like behavior.

It has been also demonstrated that BBB integrity disruption is partially mediated by TNFα, and thus, it has been hypothesized that this factor contributes to blockade of the recovery from prolonged depression-like behavior [70]. An increased level of TNFα in non-recovered mice displaying depression-like behavior is accompanied by greater hippocampal activation of GSK3, higher levels of interleukin-17A and -23, and lower level of the BBB tight junction proteins in comparison to recovered and control animals [70]. The administration of TDZD-8 reduces inflammatory cytokines levels, increases tight junction proteins level, and reverses impaired recovery from depression-like behavior. Similar can be observed when a TNFα inhibitor, etanercept, is administrated. These observations indicate that the stress-induced GSK3 activation contributes to the disruption of BBB integrity mediated by pro-inflammatory factors, particularly TNFα [70].

It is worth noting that multiple research groups have reported that the manipulation of the gastrointestinal tract microbiome status affects anxiety- and depressive-like behaviors in rodents, and that the administration of probiotics reduces such behaviors [151]. The gut microbiota play an active role in immunity and inflammation [152]. However, it cannot be excluded that GSK3β-regulated inflammatory reaction within the peripheral immune system is the place of origin of the inflammation-induced depression.

Thus, it can be concluded, that the deregulation of molecular pathways and cellular processes, such as neurotrophic factors-regulated, β-catenin-mediated, and inflammatory pathways, miRNAs expression, and DNA modification, observed during stress-induced conditions can be directly or indirectly linked with the malfunctioning of GSK3β.

## 4. DA and 5-HT/AKT/GSK3 Pathway Modulation and Its Behavioral Consequences

DA regulates AKT/GSK3 pathway mainly in the β-arrestin 2-mediated manner. Urs et al. demonstrated that in mice, GSK3β knockout in D2R-expressing neurons, but not in D1R-expressing cells, mimics the action of antipsychotics [153]. The stabilization of β-catenin, a downstream target of GSK3β, in D2R-positive neurons, does not affect mice behavior, which suggests that in this context, GSK3β does not act through the β-catenin-mediated pathway [153]. Constant hyper-dopaminergy in mice lacking DA transporter (DAT), which removes DA from the synaptic cleft, leads to a reduction in AKT activity and an increased activity of GSK3α/β [154]. Additionally, the administration of DA receptor agonists, such as amphetamine, methamphetamine, or apomorphine, to normal mice results in AKT inhibition [155]. The depletion of DA has the opposite effect [156]. Hyper-dopaminergic mice display concomitant novelty-induced locomotor hyperactivity [157], which could be reduced by GSK3 inhibitors in the DAT lacking mice, and in amphetamine-treated normal mice [158,159]. Additionally, the inhibitory Ser9-phosphorylation of GSK3β is decreased in murine medial PFC after exposure of animals to novel objects, but the DAT knockdown mice exhibit no such decrease [160]. It has been found that the deletion of D3R in DAT knockdown mice restores novelty-induced GSK3β activation in the medial PFC. Moreover, inhibition or knockdown of GSK3β, but not the α isozyme, in the medial PFC of wild-type mice impairs recognition memory [160], which suggests that in the medial PFC, D3R acts via GSK3β signaling to play a role in the novel objects recognition memory.

It has been demonstrated that GSK3β^+/−^ heterozygotic mice are less responsive to amphetamine [158], whereas mice expressing a constitutively active GSK3β mutant develop a locomotor hyperactivity phenotype recapitulating the hyper-dopaminergy conditions [161]. 

The AKT/GSK3β pathway is also affected by 5-HT signaling. As has already been mentioned, 5-HT receptors play antagonistic roles in the regulation of GSK3β activity [13]. Loss-of-function mutation of tryptophan hydroxylase 2, a rate-limiting enzyme of neuronal 5-HT synthesis, causes a severe 5-HT deficiency and results in an increase in GSK3 activity in the frontal cortex [162]. It is accompanied by behavioral abnormalities in tests assessing 5-HT-mediated emotional states, such as anxiety and aggression, which can be reversed by the administration of a selective GSK3β inhibitor, TDZD-8 [162].

The presented data clearly demonstrate that the action of DA and 5-HT receptors is mediated by GSK3β and that the inhibition of the kinase can restore the effects of hyper-dopaminergy and hypo-serotoninergy.

## 5. Influence of Anti-Depressants on GSK3β Activity

A wide spectrum of currently used therapeutic agents in MDD treatment originates from the monoaminergic hypothesis of depression pathogenesis. However, the anti-depressant-like action of GSK3β inhibitors strongly suggests the involvement of this kinase in the pathogenesis of this disorder. Medicaments used in MDD treatment are summarized in Table 1.

### 5.1. Tricyclic Anti-Depressants

Members of the Tricyclic anti-depressants (TCA) group are nonspecific agents acting on a wide range of neuronal receptors and monoamine transporters. The first described TCA is imipramine, which increases Ser9 phosphorylation of GSK3β in the mouse cerebral cortex, hippocampus, and striatum [163]. It is hypothesized that this effect is caused by the high affinity of imipramine binding to SERT, NA transporter (NET), and 5-HT_2A_R (inhibition constants (Ki) are 1.3 nM, 20–37 nM [164], and 80–150 nM [165], respectively). NA acting through PKA and PKC [36,38], and 5-HT acting via receptors other than 5-HT_2A_R inactivate GSK3β, whereas the blockade of 5-HT_2A_R prevents the inhibition of AKT [13]. A major metabolite of imipramine is desipramine, which has the highest affinity to NET (Ki 0.63–3.5 nM [164]). One week of desipramine administration significantly lowers the expression of GSK3β gene in the mouse frontal cortex [166].

Another TCA, amitriptyline, which blocks SERT and NET (Ki 2.8–4.3 nM and 19–35 nM [164], respectively), enhances phosphorylation of the GSK3β Ser9 residue [167]. Moreover, amitriptyline acts as an agonist of TrkB in the absence of its ligands [168]. Activated TrkB regulates GSK3β via PI3K/AKT pathway [91]. 

To the best of our knowledge, an action of doxepin has not been studied in the context of GSK3β activity. However, it is known that chronic administration of doxepin does not alter AKT gene expression in rodent hippocampus [169,170]. On the other hand, in the rat model of Alzheimer’s disease, doxepin reverses the effect of soluble amyloid β 1-42-induced memory impairment acting through the PI3K/AKT/mTOR signaling pathway [171]. 

Among other TCAs, opipramol has the highest affinity to σ opioid receptors (Ki for σ1 receptor is about 0.2–50 nM [172]) and acts as their agonist [173]. There is no literature data on the influence of opipramol on GSK3β activity. However, dehydroepiandrosterone, the most abundant neurosterol in the central nervous system, improves cognitive function, ameliorates depressive-like behaviors, and stimulates neurogenesis in the dentate gyrus, increasing the activity of AKT and phosphorylation of the GSK3β Ser9 residue via σ1 receptors [174].

### 5.2. Selective Serotonin Reuptake Inhibitors

Therapeutic agents, such as citalopram, escitalopram, fluoxetine, fluvoxamine, paroxetine, and sertraline, belong to selective serotonin reuptake inhibitors (SSRIs). They increase concentrations of 5-HT in the synaptic cleft by inhibiting SERT. They also act as agonists and antagonists of σ1 receptors (only sertraline is a σ1 receptor antagonist) [175].

Fluoxetine increases the inhibitory phosphorylation of GSK3β in mice hippocampi, and the anti-depressant effect of fluoxetine is diminished when the phosphorylation is hindered [176]. Citalopram and its (S)-stereoisomer escitalopram both stimulate the inhibitory phosphorylation of GSK3β [177,178]. Furthermore, escitalopram upregulates AKT activity by Ser473 phosphorylation [178]. The same effect as fluvoxamine [179], whereas paroxetine inhibits GSK3β via functional synergism with FK506 binding protein 51 [180]. Additionally, SSRIs regulate BDNF gene expression, which could explain a requirement of several weeks of treatment to achieve the full therapeutic effect. Coppell et al. analyzed BDNF transcripts in rat hippocampi, after acute and chronic (14 days) administration of fluoxetine, paroxetine, and sertraline. They have found that 4 h after acute SSRI administration (the single injection) and 4 h after the last injection of the chronic treatment, BDNF gene expression level was significantly reduced in comparison to respective controls. Moreover, 24 h after the single injection, the BDNF expression level did not differ from control, whereas 24 h after the last injection of the chronic treatment, the expression level of BDNF was significantly higher than in control [181]. As BDNF is a major contributor to GSK3β activity regulation [91], factors which regulate its expression also influence GSK3 action.

The most elusive mechanism of action is attributed to sertraline. Besides its anti-depressant effect, sertraline has antiproliferative potential. In melanoma cells, sertraline downregulates AKT upregulating, thus GSK3 activity [182]. Additionally, in human breast adenocarcinoma, sertraline stimulates overexpression of REDD1 and inhibits mTOR [183]. Both AKT inhibition and REDD1 overexpression have been observed in depression [53,101]. However, in contrast to the depression-like effect of sertraline in cancer cells, its anti-depressant properties could be explained by the high affinity of sertraline to SERT (Ki is about 0.4 nM [164]).

### 5.3. Selective Serotonin and Noradrenaline Reuptake Inhibitors

Selective serotonin and noradrenaline reuptake inhibitors (SSNRIs) increase amounts of 5-HT and NA in synaptic clefts by blocking SERT and NET. Members of the SSNRIs group, atomoxetine and milnacipran, induce AKT by increasing its activatory phosphorylation [184,185], whereas duloxetine stimulates expression of AKT and inhibits expression of both GSK3 isozymes [186].

### 5.4. α2-Receptor Blockers

Members of this group, mianserin and mirtazapine, besides their anti-adrenergic activity, can also block histaminergic H1 receptors and a wide spectrum of 5-HT receptors. Additionally, mianserin is a NET inhibitor [187]. Mirtazapine enhances BDNF gene expression in the cerebral cortex and hippocampus. It has been hypothesized that its effect is due to the activation of the PI3K/AKT pathway and inhibition of GSK3β [188]. Furthermore, the anti-depressant effect of mianserin and mirtazapine can be attributed to the downregulation of 5-HT_2A_R expression [189], which activates GSK3β via inhibition of AKT [190].

### 5.5. Monoamine Oxidase Inhibitors

Monoamine oxidase inhibitors (MAOi) reversibly or irreversibly inhibit the activity of both isozymes of monoamine oxidase, A and B, thus increasing concentrations of 5-HT, adrenaline, NA, melatonin, and DA in the synaptic cleft. Tranylcypromine, a nonselective MAO-A and MAO-B inhibitor, enhances the activity of AKT and stimulates the expression of BDNF [191,192]. The elevated BDNF expression in the hippocampus, has also been demonstrated in moclobemide treatment [193]. Unfortunately, there are no data reporting a direct influence of MAOi on GSK3β activity.

### 5.6. Selective Noradrenaline Reuptake Inhibitors 

Among many selective NET inhibitors, only reboxetine has been linked to GSK3β activity. It stimulates the overexpression of BDNF and the upregulation of AKT [192].

### 5.7. Selective Noradrenaline and Dopamine Reuptake Inhibitors 

Inhibitors of NET and DAT are widely used in the treatment of depression, Parkinson’s disease, and attention deficit hyperactivity disorder. However, only methylphenidate has been analyzed in the context of GSK3β activity. A chronic administration of methylphenidate increases activating phosphorylation of AKT and inhibits GSK3β in the cerebral cortex in mice [194].

### 5.8. Melatonin Receptors Agonists 

Melatonin receptors are G-protein coupled receptors, which, in humans, occur in two types: MR1 and MR2. Their action is associated with modulation of PKA, PKC, and PLC activity. Melatonin increases the amount of Ser473-phosphorylated AKT and Ser9-phosphorylated GSK3β [195]. There are some melatonin receptors agonists, including ramelteon, agomelatine, and tasimelteon. Among them, agomelatine is suggested to stimulate the release of neuroprotective agents, such as BDNF, and act on the ERK/AKT/GSK3β signaling pathway. Agomelatine also acts as an antagonist of 5-HT_2C_R [196].

### 5.9. Trazodone

The action of trazodone is complex. Besides its anti-adrenergic and anti-histaminic activity, trazodone is an agonist of 5-HT_1A_R and an antagonist of 5-HT_2A_R. Moreover, trazodone acts as a weak inhibitor of SERT [197]. Trazodone slightly elevates the concentration of 5-HT in synaptic clefts and promotes AKT activity and GSK3β inactivation [197].

### 5.10. Lithium

Salts of lithium are used in the treatment of bipolar disorder and MDD. Lithium inhibits both isozymes of GSK3 directly [198] by competition with magnesium ions [199]. Lithium can also activate AKT and, thus, inhibit GSK3 indirectly [158,200,201]. The mechanism of the lithium-mediated AKT activation relies on the destabilization of the β-arrestin 2/AKT/PP2A complex in which AKT is dephosphorylated and inactivated [202].

### 5.11. Ketamine

Ketamine is an anesthetic that, in subanesthetic doses, demonstrates rapid-onset efficacy in patients with severe and treatment-refractory depression. Ketamine has a wide spectrum of effects, both immediate and delayed, on neuronal function, but its anti-depressant mechanism of action has not been well characterized yet. Ketamine can block the activity of NMDA receptors [203]. However, it also enhances glutamatergic transmission in the PFC and hippocampus [204]. It has been hypothesized that this paradox is a result of a much higher affinity of ketamine to NMDAR in GABAergic interneurons than to NMDAR in pyramidal neurons. The consequent ketamine-induced inhibition of the interneurons results in disinhibition of the pyramidal cells [205]. It might explain why, in high anesthetic doses, ketamine causes depression of the central nervous system, whereas its subanesthetic anti-depressant doses act depressively on inhibitory interneurons. Ketamine treatment leads to increased excitability of disinhibited glutamatergic neurons and may stimulate BDNF release [206]. The ketamine-induced BDNF release results in the activation of MAPK and ERK [207], downstream effectors of TrkB signaling. Ketamine also rapidly elevates the level of active AKT [204]. Both the TrkB/MEK/ERK and TrkB/PI3K/AKT pathways can activate mTOR and inactivate GSK3β [13,204]. The activation of the former signaling pathway leads to an enhanced transcription of synaptic proteins, whereas stimulation of the latter pathway prevents internalization of AMPARs and shrinking of the synapse by the reduction in GSK3β-mediated PSD95 phosphorylation [208]. It has been shown that inhibition of GSK3 is necessary for the rapid anti-depressant action of ketamine in mice [209], whereas knock-in mice with constitutively active GSK3β are insensitive to the anti-depressant action of ketamine [208].

### 5.12. Electroconvulsive Therapy

This kind of therapy is still widely used in the treatment of MDD, mania, and catatonia, especially in their treatment-resistant forms. Interestingly, acute electroconvulsive therapy has been found to increase the level of the Ser9-phosphorylated GSK3β in the murine frontal cortex and hippocampus [210].

Summarizing, it has been demonstrated that different classes of therapeutic used in the treatment of MDD act through the GSK3β pathway or affect the activity of the kinase directly or indirectly.

## 6. Conclusions

GSK3β activity and expression in a cell are regulated by a wide spectrum of neurotransmitters, neuromodulators, and neurotrophic factors. Many of them are targets of anti-depressant treatments. Thus, it is not surprising that such treatments also influence the activity of GSK3β. Despite a variety of hypotheses considering potential factors and risks in the pathogenesis of MDD, nearly all of the hypotheses can be considered in the context of the GSK3β activity, which places the kinase at a central point of depression development and treatment. Abnormal GSK3β activity, an altered profile of its expression, and genetic polymorphism correlate with the MDD pathogenesis, age of its onset, and severity. However, further studies are needed to fully elucidate if the elevated activity of GSK3β is a reason or an effect of mood disorders.

## Figures and Tables

**Figure 1 cells-09-00727-f001:**
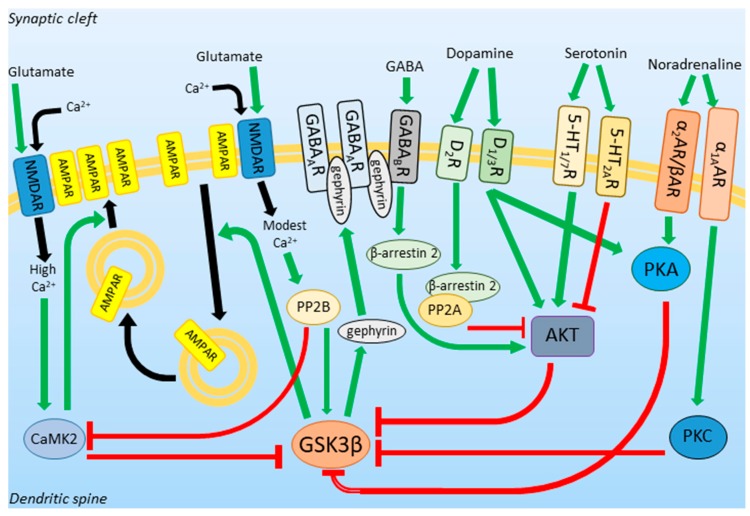
Neurotransmitters acting through protein kinases and phosphatases regulate glycogen synthase kinase 3β (GSK3β) activity. GSK3β influences the strength of excitatory and inhibitory synapses. Green arrows represent activation, red arrows represent inhibition, black arrows represent translocation. Abbreviations: NMDAR: *N*-methyl-d-aspartate receptor, AMPAR: α-amino-3-hydroxy-5-methyl-4-izoxazole propionic acid receptor, GABA_B_R: γ-aminobutyric acid receptor type B, D_1/2/3_R: dopamine receptors, 5-HT_1/2A/7_R: serotonin receptors, α_1A_/α_2_/βAR: adrenergic receptors, PKA: protein kinase A, AKT: protein kinase B, PKC: protein kinase C, PP2A/2B: protein phosphatases 2A/2B, CaMK2: calcium/calmodulin dependent protein kinase 2, GSK3β: glycogen synthase kinase 3β.

**Figure 2 cells-09-00727-f002:**
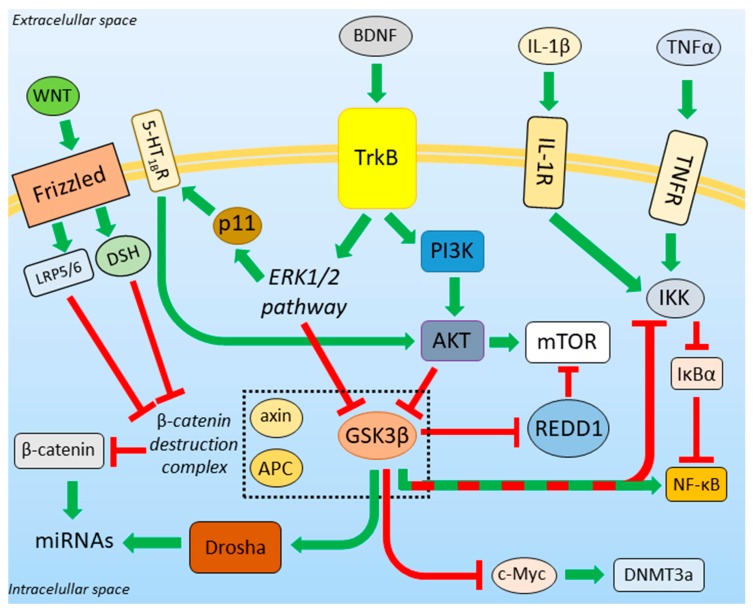
GSK3β is a part of neurotrophic and pro-inflammatory factors signaling pathways. Green arrows represent activation, red arrows represent inhibition. Green-red arrow denotes a dual, dependent on the physiological state of a cell, action of GSK3β. Abbreviations: WNT: Wingless-related integration site, LRP5/6: low-density lipoprotein receptor-related protein 5 and 6, DSH: Disheveled, 5-HT_1B_R—serotonin 1B receptor, ERK: extracellular signal-regulated kinase, APC: adenomatous polyposis coli, BDNF: brain-derived neurotrophic factor, TrkB: tropomyosin receptor kinase B, PI3K: phosphoinositide 3-kinase, AKT: protein kinase B, IL-1β: interleukin 1β, IL-1R: interleukin 1β receptor, TNFα: tumor necrosis factor α, TNFR: tumor necrosis factor α receptor, IKK: IκB kinase, IκBα: nuclear factor of light polypeptide gene enhancer in B-cells inhibitor α, NF-κB: Nuclear factor κ-light-chain-enhancer of activated B cells, mTOR: mammalian target of rapamycin, REDD1: regulated in development and DNA damage response-1, DNMT3a: DNA methyltransferase 3a, GSK3β: glycogen synthase kinase 3β.

**Table 1 cells-09-00727-t001:** Main classes of medications currently used in major depressive disorder (MDD) treatment, with examples of their members and targets of their action. Trazodone, ketamine, and lithium do not belong to any of the listed groups.

Anti-Depressants Class	Class Members	Targets	Effects on GSK3β Pathway
Tricyclic anti-depressants (TCA)	Imipramine, desipramine, clomipramine, amitriptyline, protriptyline, doxepin, dosulepin, opipramol	Nonspecific Serotonin transporter inhibitors, noradrenaline transporter inhibitors, anti-serotoninergic, D_2_R blockers, anti-cholinergic, anti-adrenergic, anti-histaminic, sigma receptors agonists/antagonists	Prevent AKT inhibition through blockade of 5-HT_2A_R, enhance GSK3β inhibition via PKA, PKC, TrkB/PI3K/AKT, and σ1R
Selective serotonin reuptake inhibitors (SSRI)	Citalopram, escitalopram, fluoxetine, fluvoxamine, paroxetine, sertraline	Mainly serotonin transporter inhibitors, sigma receptors agonists/antagonists	Enhance GSK3β inhibition via 5-HT_1/7_R/PI3K/AKT, upregulate BDNF expression
Selective serotonin and noradrenaline reuptake inhibitors (SSNRI)	Atomoxetine, desvenlafaxine, duloxetine, levomilnacipran, milnacipran, sibutramine, tramadol, venlafaxine	Serotonin and noradrenaline transporters inhibitors	Increase AKT activity and expression, downregulate GSK3α/β expression
α_2_-receptor blockers	Mianserin, mirtazapine	Anti-adrenergic, anti-serotoninergic, anti-histaminic, noradrenaline transporter inhibitors	Upregulate BDNF expression, downregulate 5-HT_2A_R expression, activate PI3K/AKT/GSK3β
Monoamine oxidase inhibitors (MAOi)	Isocarboxazid, tranylcypromine, moclobemide, toloxatone, rasagiline, selegiline	Reversible or irreversible inhibition of MAO-A and MAO-B	Upregulate BDNF expression, increase AKT activity
Selective noradrenaline reuptake inhibitors (SNRI)	Reboxetine, viloxazine, maprotiline	Noradrenaline transporter inhibitors	Upregulate BDNF expression, increase AKT activity
Selective noradrenaline and dopamine reuptake inhibitors (SNDRI)	Amineptine, bupropion, dexmethylphenidate, methylphenidate, phenylpiracetam	Noradrenaline and dopamine transporters inhibitors	Increase AKT activity, enhance GSK3β inhibition
Melatonin receptor agonists	Ramelteon, agomelatine, tasimelteon	Activate melatonin receptors	Increase AKT activity, enhance GSK3β inhibition, stimulate BDNF release
	Trazodone	5-HTR agonist/antagonist, serotonin transporter inhibitor, anti-adrenergic, anti-histaminic	Enhance GSK3β inhibition via 5-HT_1A_R stimulation and 5-HT_2A_R blockade
	Lithium	A lot of targets Directly activates AKT, directly inhibits GSK3α/β, destabilizes β-arrestin 2/AKT/PP2A complex	Increase AKT activity, enhance GSK3β inhibition
	Ketamine	A lot of targets NMDAR antagonist, indirect agonist of AMPAR, opioid receptors antagonist, D_2_R agonist, sigma receptors agonist, anti-cholinergic, cholinesterase inhibitor, 5-HT/NA/DA reuptake inhibitor, blocker of voltage-dependent sodium and calcium channels, nitric oxide synthase inhibitor	Enhance GSK3β inhibition via TrkB/MEK/ERK and TrkB/PI3K/AKT

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
