# Peer review of "GSK3β: A Master Player in Depressive Disorder Pathogenesis and Treatment Responsiveness"

_cells, 2020, doi:10.3390/cells9030727_

Round 1

Reviewer 1 Report

Evidence shows that GSK3 contributes to pathological processes in a wide range of psychiatric and neurological disorders including major depressive disorder (MDD). In the submitted article authors discuss how different pathways/mechanisms affect GSK3β and thus may contribute to the pathogenesis of MDD and how different antipsychotic drugs affect GSK3β expression/activity.

1. What is missing in the submitted article, in my opinion, is the evidence from animal models showing that genetically reducing or pharmacologically inhibiting GSK3β can ameliorate symptoms of depression in different behavioural paradigms. Thus, discussing several findings from GSK3 transgenic and knockout mice would substantiate the submitted article. For example, double GSK3α/β knockin mice show increased susceptibility to the development of learned helplessness (Polter et al, 2010). Pardo et al, 2016 used GSK3α knockin mice and GSK3β knockin mice to determine the contribution of either GSK3 isozyme to the learned helplessness model of depression-like behaviour. Moreover, reduction of GSK3β levels in heterozygous GSK3β+/− knockout mice decreases immobility in the forced swim test (O’Brien et al, 2004) and the tail suspension test (Beaulieu et al, 2008), which are indicative of depression-like behaviour. Transgenic expression of GSK3β in mouse brain rescued lithium-sensitive immobility in the forced swim test (O’Brien et al, 2011). Moreover, GSK3β silencing in the dentate gyrus produces antidepressant-like effects in stressed mice (Omata et al, 2011). In terms of pharmacological interventions would be worth to write about the rapid antidepressive-like activity of specific GSK3 inhibitor - L803-mts (Kaidanovich-Beilin et al, 2004) and other inhibitors: SB 216763 (Liu et al, 2013) and TDZD-8 (Cheng et al, 2018) producing anti-depressant effects.  

2. Chapter 1.4. In my opinion “to the best of our knowledge” is an overstatement, because there is one report, one but still, by Li et al, 2014 on GSK3β in the ventral tegmental area (VTA) in cocaine-induced conditioned place preference. Because VTA is a part of mesolimbic system which is a reward pathway and the motivation for pleasure is lost in depression, it would be interesting to discuss this work and other works on GSK3β in controlling motivation for reward, for example, Miller et al, 2014. In line with this, it would be interesting to discuss the mesolimbic system in which dopamine neurons from VTA project to nucleus accumbens (NAc). Silencing of GSK3β in the NAc shell increases depression-related behaviour (Crofton et al, 2017). Overexpression of GSK3β in the NAc induces prodepression-like effects in forced swim test after social defeat stress (Wilkinson et al, 2011). While these studies are contradictory, Crofton et al, 2017 suggested that different stress exposure may play a role.

3. Chapter 1.11. This chapter is missing the entire Eleonore Beurel’s work on GSK3β in neuroinflammation in relation to depression: Beurel et al, 2009, 2013; Cheng et al, 2016 and Cheng et al, 2018. The latter study may be of importance as it depicts GSK3β as a central player in the disruption of BBB integrity in relation to TNFα and inflammation in depression - issues that are raised in this chapter.

4. Chapter 2 could mention findings from mice in which GSK3β is deleted in dopamine receptor-positive neurons: D1 (Urs et al, 2012), D2 (Urs et al, 2012; Li et al, 2019), or from dopamine transporter knockdown mice and D3 receptor regulation of GSK3β (Chang et al, 2020).

5. Chapter 3.10. Given the overwhelming literature on lithium in the treatment of bipolar disorder and MDD, the chapter on lithium looks rather modest.

6. The single chapters, for example, 1.6 and others could be single sentence concluded at the end.

7. Line 46: Authors mention that there are “two isozymes of GSK3”, later they use “GSK3 isoforms”

8. Line 331: “GSK3β heterozygotic mice” clarify that this is heterozygous loss

9. Some sentences are missing a reference or references could be different:
Line 102: ref 27 could be different
Line 110: sentence is missing ref
Line 127: in ref 40 Alltoa et al, 2010 found upregulated GSK3β gene expression, while the sentence deals with GSK3β activity
Line 240: ref 59 could be different

10. There are few typos and errors:
Line 102: gephiryn -> gephyrin;
Line 173: synaptic spines -> dendritic spines;
Line 177: amount -> number;
Line 189: both these -> both of these or both pathways;
Line 202: associated to -> associated with;
Line 225 and 230: apoptiotic -> apoptotic;
Line 231: do not affect neither -> affect neither;
Line 235: so-called -> the so-called;
Line 258: required to -> required for;
Line 270: increased levels -> increased numbers

Author Response

  “1. What is missing in the submitted article, in my opinion, is the evidence from animal models showing that genetically reducing or pharmacologically inhibiting GSK3β can ameliorate symptoms of depression in different behavioural paradigms. Thus, discussing several findings from GSK3 transgenic and knockout mice would substantiate the submitted article. For example, double GSK3α/β knockin mice show increased susceptibility to the development of learned helplessness (Polter et al, 2010). Pardo et al, 2016 used GSK3α knockin mice and GSK3β knockin mice to determine the contribution of either GSK3 isozyme to the learned helplessness model of depression-like behaviour. Moreover, reduction of GSK3β levels in heterozygous GSK3β+/− knockout mice decreases immobility in the forced swim test (O’Brien et al, 2004) and the tail suspension test (Beaulieu et al, 2008), which are indicative of depression-like behaviour. Transgenic expression of GSK3β in mouse brain rescued lithium-sensitive immobility in the forced swim test (O’Brien et al, 2011). Moreover, GSK3β silencing in the dentate gyrus produces antidepressant-like effects in stressed mice (Omata et al, 2011). In terms of pharmacological interventions would be worth to write about the rapid antidepressive-like activity of specific GSK3 inhibitor - L803-mts (Kaidanovich-Beilin et al, 2004) and other inhibitors: SB 216763 (Liu et al, 2013) and TDZD-8 (Cheng et al, 2018) producing anti-depressant effects. “ 

We described studies on GS3β in the animal models of depression, and on the inhibitors of GS3β used to treat depression in two new subchapters (lines 209-252):

"3.1.  GSK3β in the animal models of depression

In 1949, Cade suggested that lithium, the classical mood stabilizer, might be a possible therapeutic for psychiatric diseases based on its behavioral effects in guinea pigs [1]. Since then, the rodent behavioral characteristics have become the high value models in mood disorders treatment studies [2]. The discovery that GSK3β is a direct target of lithium action [3] has raised the possibility that lithium exerts its effects through modulating activity of the kinase. To identify the mood-altering role of GSK3 α and β isozymes molecular methods have been employed in various animal models of depression.

On the one hand, behavioral characteristics of GSK3α/β21A/21A/9A/9A knock-in mice (with serine to alanine mutations to block the inhibitory phosphorylation of serine 21 and 9 in GSK3 ɑ and β, respectively) demonstrated that the animals exhibited a heightened response to a novel environment and that administration of amphetamine caused over 2.5-fold greater hyperactivity compared to control mice [4]. It emphasized the importance of GSK3α/β activity in development of manic-like disturbances. On the other hand, the knock-in animals showed an increased vulnerability to stress-induced depressive-like behavior in the learned helplessness, forced swim, and tail suspension tests [4]. Additionally, the lack of adaptability of these knock-in mice to stress involved also anxiety, which often coexists with depression. The animals displayed a mild-anxious behavior in the elevated plus maze [4]. Later, using GSK3β knock-in mice, it has been established that increased activity of β isozyme of GSK3 is sufficient to impair mood regulation in learned helplessness model of depressive-like behavior, whereas GSK3α activity alone does not impair this process [5].

Bilateral intra-hippocampal injections of lentivirus expressing shRNA anti-Gsk3β induce antidepressant-like effects in chronically stressed mice, in the forced swim and tail suspension tests [6]. Moreover, heterozygous loss of Gsk3β causes behavioral defects that mimics the action of lithium [7], whereas transgenic expression of Gsk3β in Gsk3β+/- heterozygotes reverses these defects [8]. Additionally, the same effect is observed when GSK3β is overexpressed in lithium-treated mice [8]. Rescue of the heterozygous loss of Gsk3β or lithium-induced phenotype by restoring the activity of GSK3β strongly support the hypothesis that the phenotype is due to specific inhibition of the kinase.

3.2.  Inhibitors of GSK3β in depression

Several lines of evidence has shown that GSK3β contributes to development of such prevalent diseases as diabetes, Alzheimer’s disease, as well as mood disorders. GSK3β inhibitors can be classified into three categories: non-ATP-competitive, ATP-competitive, and substrate competitive inhibitors [9].

It has been demonstrated that non-ATP-competitive GSK3β inhibitors ameliorate depressive-like behavior in rodents. It has been also shown that prolonged learned helplessness is reversible and is maintained by abnormally active GSK3, whereas treatment with TDZD-8 reverses the impaired recovery from learned helplessness [10]. In turn, an ATP-competitive GSK3β inhibitor SB216763 has been found to increase anti-depressant responses in the forced swim test [11], whereas SAR502250 improved the stress-induced physical state in the chronic mild stress test in mice [12].

Intracerebroventricular injection of a GSK3β substrate competitive inhibitor, L803-mts, has reduced duration of immobility in the forced swim test in mice, in comparison to control animals [13]. Additionally, the expression level of β-catenin, a substrate of GSK3β, was increased in the inhibitor-treated animals [13]."

The above studies demonstrate that GSK3β inhibitors produce anti-depressive-like behavior, and suggest the potential of the kinase inhibitors as anti-depressants.

“2. Chapter 1.4. In my opinion “to the best of our knowledge” is an overstatement, because there is one report, one but still, by Li et al, 2014 on GSK3β in the ventral tegmental area (VTA) in cocaine-induced conditioned place preference. Because VTA is a part of mesolimbic system which is a reward pathway and the motivation for pleasure is lost in depression, it would be interesting to discuss this work and other works on GSK3β in controlling motivation for reward, for example, Miller et al, 2014. In line with this, it would be interesting to discuss the mesolimbic system in which dopamine neurons from VTA project to nucleus accumbens (NAc). Silencing of GSK3β in the NAc shell increases depression-related behaviour (Crofton et al, 2017). Overexpression of GSK3β in the NAc induces prodepression-like effects in forced swim test after social defeat stress (Wilkinson et al, 2011). While these studies are contradictory, Crofton et al, 2017 suggested that different stress exposure may play a role.”

We described studies on the role of GSK3β in the ventral tegmental area and in the control of motivation in Chapter 2, lines 146-165:

"Anhedonia and loss of motivation are one of the main MDD symptoms. They are caused by a disrupted interplay between different brain structures, such as ventral tegmental area, nucleus accumbens and cingulate cortex, which can be grouped in the so-called reward circuit [45]. Wilkinson et al. have demonstrated that in the nucleus accumbens, a key region of the circuit, the amount of Ser9-phosphorylated GSK3β is downregulated in the mouse social defeat model of depression [46]. The increased activity of the kinase can be observed in susceptible, but not in resilient animals. Additionally, a similar pro-depression-like effect can be induced by GSK3β overexpression in the nucleus accumbens, while the expression of inactive GSK3β mutant promotes resilience to social defeat stress [46]. On the other hand, Crofton et al. have found that the knock-down of GSK3β in the nucleus accumbens shell increases cocaine self-administration and depression-like behavior in the social contact test in rats [47, 48].  This discrepancy between stressor type and its effect within the same structure is also reflected in the activity of ventral tegmental area – a midbrain structure which delivers input to nucleus accumbens.  It has been shown that chronic social defeat stress enhances the phasic firing rate of ventral tegmental area neurons in defeated rodents [49], however, chronic stress can also lead to atrophy of ventral tegmental area system [50]. It has been hypothesized that these opposing outcomes might be due to a different nature of the stressors, heterogeneity of ventral tegmental area cells, or the time of day when the experiments were performed [51]. Interestingly, it has been found that the level of Ser9-phosphorylated GSK3β in ventral tegmental area and other reward-related brain structures in naive rats shows a significant circadian rhythmicity [52]."

“3. Chapter 1.11. This chapter is missing the entire Eleonore Beurel’s work on GSK3β in neuroinflammation in relation to depression: Beurel et al, 2009, 2013; Cheng et al, 2016 and Cheng et al, 2018. The latter study may be of importance as it depicts GSK3β as a central player in the disruption of BBB integrity in relation to TNFα and inflammation in depression - issues that are raised in this chapter.”

We added information about studies on neuroinflammation to the Chapter 3.8:

-          lines 386-390:

"Additionally, it has been shown that IL-6 induces production of inflammatory T helper 17 cells (Th17) increasing thus levels of these cells in brain during depression-like states [14]. Moreover, it has been shown that administration of Th17 cells promotes depression-like behaviors in mice, and inhibition of production and functioning of Th17 cells reduces vulnerability of the animals to depression-like behavior [14]."

-          lines 399-401:

"Expression of IL-6 is regulated by a transcription factor STAT3 [18]. GSK3β promotes STAT3 activation and thus, stimulates the expression of IL-6. Inhibition and knock-down of GSK3β, but not GSK3α, strongly inhibits IL-6 production by glial cells both in vitro and in vivo [18]."

-          lines 412-427:

"Elevation of cytokines and chemokines levels in hippocampi of mice displaying depression-like behavior is mediated by Toll-like receptor 4 (TLR4) activity [21]. It has been shown that learned helplessness paradigm activates GSK3 in a wild-type mouse hippocampus, but not in TLR4 knock-out mice [21]. Additionally, TDZD-8 attenuates an increased activation of NF-κB upon TLR4 stimulation [21], what indicates that GSK3 mediates a TLR4-related pro-inflammatory reaction associated with depression-like behavior. 

It has been also demonstrated that BBB integrity disruption is partially mediated by TNFα, and  thus, it has been hypothesized that this factor contributes to blockade of the recovery from prolonged depression-like behavior [10]. An increased level of TNFα in non-recovered mice displaying depression-like behavior is accompanied by greater hippocampal activation of GSK3, higher levels of interleukin-17A and -23, and lower level of the BBB tight junction proteins in comparison to recovered and control animals [10]. Administration of TDZD-8 reduces inflammatory cytokines levels, increased tight junction proteins levels and reverses impaired recovery from depression-like behavior. The similar can be observed when a TNFα inhibitor, etanercept, is administered. These observations indicate that the stress-induced GSK3 activation contributes to the disruption of BBB integrity mediated by pro-inflammatory factors, particularly TNFα [10]."

“4. Chapter 2 could mention findings from mice in which GSK3β is deleted in dopamine receptor-positive neurons: D1 (Urs et al, 2012), D2 (Urs et al, 2012; Li et al, 2019), or from dopamine transporter knockdown mice and D3 receptor regulation of GSK3β (Chang et al, 2020).

Information about studies on the effect of GSK3β deletion in dopaminergic neurons is added to the Chapter 4:

-          lines 441-445:

"Urs et al. have demonstrated that in mice, GSK3β knock-out in D2R-expressing neurons, but not in D1R-expressing cells, mimics an action of anti-psychotics [22]. Stabilization of β-catenin, a downstream target of GSK3β, in D2R-positive neurons does not affect mice behavior, what suggests that in this context, GSK3β does not act through β-catenin-mediated pathway [22]."

-          lines 452-457:

"Additionally, the inhibitory Ser9-phosphorylation of GSK3β is decreased in murine medial PFC after exposure of animals to novel objects, but the DAT knock-down mice exhibit no such decrease [29]. It has been found that the deletion of D3R in DAT knock-down mice restores novelty-induced GSK3β activation in the medial PFC. Moreover, inhibition or knock-down of GSK3β, but not the α isozyme, in the medial PFC of wild-type mice impairs recognition memory [29], which suggests that in the medial PFC, D3R acts via GSK3β signaling to play a role in the novel objects recognition memory."

“5. Chapter 3.10. Given the overwhelming literature on lithium in the treatment of bipolar disorder and MDD, the chapter on lithium looks rather modest.”

We agree with the Reviewer that the effects of lithium in the treatment of bipolar disorder and MDD have been intensively studied and have been described in several papers. Just because that, and because the mechanism of lithium action is well understood now, we decided to describe lithium action only very concisely in Chapter 5.10.

“6. The single chapters, for example, 1.6 and others could be single sentence concluded at the end.”

At the end of every Chapter we added short summaries.

Chapter 1 (lines 130-134):

"Concluding, GSK3β is a hub which links different molecular pathways within a cell. Activity of the kinase, affected by the action of neurotransmitters, mediates neuroplasticity and directs neurons towards synaptic potentiation or depression route. Thus, because MDD is believed to be a result of dysregulation of neurotransmitters actions in different brain regions, GSK3β can be considered as a factor engaged in the MDD pathogenesis and development."

Chapter 2 (lines 184-185):

"Summarizing, the activity of GSK3β in different brain regions is affected by stress, and haplotypes of the kinase determine a severity, age of onset and drug responsiveness in MDD."

Chapter 3 (lines 435-438):

"Thus, it can be concluded, that the deregulation of molecular pathways and cellular processes, such as neurotrophic factors-regulated, β-catenin-mediated and inflammatory pathways, miRNAs expression and DNA modification, observed during stress-induced conditions can be directly or indirectly linked with a malfunction of GSK3β."

7. Line 46: Authors mention that there are “two isozymes of GSK3”, later they use “GSK3 isoforms”

We unified the nomenclature and the term “isozyme” is now used in the entire text.

8. Line 331: “GSK3β heterozygotic mice” clarify that this is heterozygous loss

We clarified that sentence in the Chapter 4 (line 458):

"It has been demonstrated that GSK3β+/- heterozygotic mice are less responsive to amphetamine [158], whereas mice expressing a constitutively active GSK3β mutant develop a locomotor hyper-activity phenotype recapitulating the hyper-dopaminergy conditions [161]."

We also corrected all mistakes, misspellings and language clumsiness, and we clarified some information.

Reviewer 2 Report

This paper reviewed the pathways linked to glycogen synthase kinase 3B and to depressive disorders.

The review paper is interesting and gives a general picture of the molecules involved in depressive disorders.

However, as in other studies of the same kind, it is impossible to shed light, for instance, on some primary and/or secondary pathways involved in GSK-3B or on whether protein kinases and phosphatases of cell membrane can mutually increase/inhibit their activity.

Author Response

"This paper reviewed the pathways linked to glycogen synthase kinase 3B and to depressive disorders.

The review paper is interesting and gives a general picture of the molecules involved in depressive disorders.

However, as in other studies of the same kind, it is impossible to shed light, for instance, on some primary and/or secondary pathways involved in GSK-3B or on whether protein kinases and phosphatases of cell membrane can mutually increase/inhibit their activity."

We agree with the Reviewer that GSK3β is involved in an intricate network of signaling and it seems almost impossible to cover all aspects of its cellular functions. In the present paper, we tried to depict as comprehensively as we could sometimes contradictory literature data on the potential roles of the kinase in the pathogenesis of depression.

Reviewer 3 Report

The manuscript has many positive attributes and will be a valuable source of information for many researchers, especially those that may be entering the field of GSK3 in depression.  The topic is certainly interesting, but there are a few points that deserve attention:

- There is no methodology given as to how exactly the reviewed publications were chosen.

- The main treatments of MDD should be more carefully introduced and described (Eg. electroconvulsive treatment must to be included, historically MAOIs are not properly contextualized)

- The authors should differentiate and specify properly the findings found in patients and the data obtained by using animal models of depression, in some points there are mixed.

- Figures are not cited properly in the text.

- Figure legends are not completed. A big portion of the legends is include in the main text.

- Figures. The turnover of AMPAR is not clear in the figure. It would be recommended to draw each receptor separately (eg. different 5-HT and DA receptors). Additionally, GABAa receptor pathway or PDS95 should be included.

- In the main text the authors explained that modest Ca++ cause calcineurin activation, but in the figure only PPB2B is included. The text and the figure should be unified.

- The authors have been cautious throughout most of the text using the expression GSK3b activation instead of GSK3B no-inhibition, it is not the same and should be clarified.

- The authors should review the bibliography. Some reference are incorrect. For example, reference number 26 not explain anything regarding the role of PSD95 or AMPAR internalization. Please, carefully review all the references included. Additionally, some bibliography crucial for this review are missing (eg. E. Beurel, L. Song, R. Jope. Inhibition of glycogen synthase kinase-3 is necessary for the rapid antidepressant effect of ketamine in mice. Mol. Psychiatry, 16 (2011), pp. 1068-1070).

- The subsections titles, as well as its order, should be revised (eg. Subsection 1.5 must be deleted; it is empty of data regarding GSK3β).

- The section regarding BDNF-regulated action of GSL3B must be rewritten. Evidence regarding the relation between depression and neurotrophines are missing (eg. hippocampal volume alteration in patients is not mentioned, and many evidences observed in experimental animals are lacking). This must be completed including the main founds in the field.

- Table 1 shows the main classes of medications used in MDD patients. However, it would be more interesting to include the main founds regarding its effect of GSK3b pathway.

- The interpretation of the data included in the review are missing. The authors should include some mini-conclusion in each section to help the reader to understand properly the role of GSK3β (eg. Selective serotonin and noradrenaline reuptake inhibitors:  Members of SSNRIs group, atomoxetine and milnacipran, induce AKT by increasing its activatory phosphorylation, whereas duloxetine stimulates expression of AKT and inhibits expression of both GSK3 isoforms. What can it mean?)

Author Response

“The manuscript has many positive attributes and will be a valuable source of information for many researchers, especially those that may be entering the field of GSK3 in depression.  The topic is certainly interesting, but there are a few points that deserve attention:

  1. There is no methodology given as to how exactly the reviewed publications were chosen.

The articles used to prepare the current manuscript have been found in PubMed with the use of such phrases as: GSK3, glycogen synthase kinase, AKT, depression, MDD, major depressive disorder, neurotransmission, neuroplasticity, neuroprotection. No year-to-year limitations were used. It is known that some data about GSK3 in a context of depression or, generally speaking, in psychiatric disorders are contradictory. Thus, we did not use any exclusion criteria, because we wanted to discuss them all. However, our review does not contain any statistical analysis of meta-data, thus, we did not include the methodology of  papers searching in the text.

  1. The main treatments of MDD should be more carefully introduced and described (Eg. electroconvulsive treatment must to be included, historically MAOIs are not properly contextualized)”

The Introduction was re-written and we described MAOis as the first therapeutics used in treatment of MDD. Additionally, we included information about electroconvulsive therapy as a currently used procedure in MDD treatment (Subchapter 1.1.):

  • lines 28-33:

The first effective treatment for MDD was established in the 1950s, when anti-depressant effects of iproniazide and imipramine were discovered. Iproniazide, originally described as an anti-tuberculosis drug, was found to be the first monoamine oxidase inhibitor (MAOi), whereas imipramine, an anti-histamine drug, was studied as an anti-psychotic for use in patients with schizophrenia [31,32]. Later, imipramine became one of the first members of tricyclic anti-depressants (TCAs).

  • lines 42-43:

 Additionally, electroconvulsive therapy, conducted for the first time in 1938, is still widely used in a treatment of MDD, especially in its drug-refractory form [34].

  1. The authors should differentiate and specify properly the findings found in patients and the data obtained by using animal models of depression, in some points there are mixed.

The manuscript was carefully re-read and, wherever it was missing, information about the models was included.

  1. Figures are not cited properly in the text.

We amended this and Figure 1 is now cited in the subchapter 1.3.

  1. Figure legends are not completed. A big portion of the legends is include in the main text.

Figure legends briefly describe the content of figures in a context of the subchapters. Additionally, the meaning of abbreviations is given. We intentionally did not include the whole description of the schemes to avoid duplication of information from the subchapters.

  1. Figures. The turnover of AMPAR is not clear in the figure. It would be recommended to draw each receptor separately (eg. different 5-HT and DA receptors). Additionally, GABAa receptor pathway or PDS95 should be included.

We corrected Figure 1. We changed the scheme of AMPARs turnover and, additionally, GABAARs and gephyrin are also shown in the revised version of the figure. However, we believe that drawing separately each receptor subtype and adding PSD95 would impair clarity of the figure. The influence of GSK3 on AMPARs internalization is clearly indicated, and the receptors are grouped depending on a pathway they affect.

  1. In the main text the authors explained that modest Ca++ cause calcineurin activation, but in the figure only PPB2B is included. The text and the figure should be unified.

The protein phosphatase 2B (PP2B) is also known as calcineurin, and both the names are mentioned in the subchapter 1.2.

  1. The authors have been cautious throughout most of the text using the expression GSK3b activation instead of GSK3B no-inhibition, it is not the same and should be clarified.

We specified the role of phosphorylation of the Ser9 residue in GSK3β in the subchapter 1.2. - lines 55-59:

Whereas a phosphorylation of the residue Tyr216 occurs during the GSK3β translation process and results in a synthesis of the fully activated kinase, the Ser9 phosphorylation seems to be the main regulatory modification during the enzyme lifespan [40]. Ser9-phosphorylated GSK3β remains inhibited and dephosphorylation of the residue results in the disinhibition (activation) of the kinase.

  1. The authors should review the bibliography. Some reference are incorrect. For example, reference number 26 not explain anything regarding the role of PSD95 or AMPAR internalization. Please, carefully review all the references included. Additionally, some bibliography crucial for this review are missing (eg. E. Beurel, L. Song, R. Jope. Inhibition of glycogen synthase kinase-3 is necessary for the rapid antidepressant effect of ketamine in mice. Mol. Psychiatry, 16 (2011), pp. 1068-1070).

We corrected bibliography and we added the missing literature. We also amended the numbering order of the cited literature to match the changes made in the manuscript.

  1. The subsections titles, as well as its order, should be revised (eg. Subsection 1.5 must be deleted; it is empty of data regarding GSK3β).

We revised the structure of the sections dividing them into new chapters and subchapters.

  1. The section regarding BDNF-regulated action of GSL3B must be rewritten. Evidence regarding the relation between depression and neurotrophines are missing (eg. hippocampal volume alteration in patients is not mentioned, and many evidences observed in experimental animals are lacking). This must be completed including the main founds in the field.

The first two paragraphs of the subchapter 3.3. provide information on the neuroanatomical findings in MDD patients and in the animal model of depression. Additionally, the BDNF level in depressed individuals in the context of the structural changes is discussed. To the revised version, we  added to the subchapter 3.3. the study of Videbech and Ravnkilde (2004) showing decreased hippocampal volumes in MDD patients (lines 258-259):

The decreased volume of hippocampus has also been found in MDD patients in comparison to healthy individuals, in an MRI-based study [45].

  1. Table 1 shows the main classes of medications used in MDD patients. However, it would be more interesting to include the main founds regarding its effect of GSK3b pathway.

We added a column to the Table 1 about the effect of anti-depressants on GSK3 and GSK3-related pathway.

  1. The interpretation of the data included in the review are missing. The authors should include some mini-conclusion in each section to help the reader to understand properly the role of GSK3β (eg. Selective serotonin and noradrenaline reuptake inhibitors:  Members of SSNRIs group, atomoxetine and milnacipran, induce AKT by increasing its activatory phosphorylation, whereas duloxetine stimulates expression of AKT and inhibits expression of both GSK3 isoforms. What can it mean?).

At the end of every Chapter, we added a short summary:

Chapter 1 (lines 130-134):

Concluding, GSK3β is a hub which links different molecular pathways within a cell. Activity of the kinase, affected by the action of neurotransmitters, mediates neuroplasticity and directs neurons towards synaptic potentiation or depression route. Thus, because MDD is believed to be a result of dysregulation of neurotransmitters actions in different brain regions, GSK3β can be considered as a factor engaged in the MDD pathogenesis and development.

Chapter 2 (lines 184-185):

Summarizing, the activity of GSK3β in different brain regions is affected by stress, and haplotypes of the kinase determine a severity, age of onset and drug responsiveness in MDD.

Chapter 3. (lines 435-438):

Thus, it can be concluded, that the deregulation of molecular pathways and cellular processes, such as neurotrophic factors-regulated, β-catenin-mediated and inflammatory pathways, miRNAs expression and DNA modification, observed during stress-induced conditions can be directly or indirectly linked with a malfunction of GSK3β.

We hope that the amendments of the manuscript will meet with the Reviewers’ approval.

Round 2

Reviewer 1 Report

I am ok with authors' response.